# Geometry-Driven Diverse and Transferable Visual Attacks on Multimodal LLMs

**Xu Zhang**
Department of Electrical and Computer Engineering
Illinois Institute of Technology
xzhang156@hawk.illinoistech.edu

**Ziqing Hu**
Perplexity AI
ziqing@perplexity.ai

**Shuo Han**
Department of Electrical and Computer Engineering
University of Illinois Chicago
hanshuo@uic.edu

**Ren Wang** [*]
Department of Electrical and Computer Engineering
Illinois Institute of Technology
rwang74@illinoistech.edu

## Abstract

Multimodal large language models (MLLMs) are increasingly vulnerable to visual jailbreak attacks, yet existing methods are often brittle, lack diversity, and transfer poorly across architectures. We revisit these limitations from a geometric perspective and introduce *Jailbreak Connectivity (JC)*, which models effective jailbreaks as connected regions of low adversarial loss rather than isolated images. By explicitly constructing continuous paths in the image space, JC generates diverse jailbreaks and exposes structural properties of multimodal vulnerabilities. We further incorporate lightweight surrogate guidance to improve cross-model transferability. Experiments on SafetyBench show that JC substantially outperforms prior methods, achieving an average attack success rate (ASR) of *79.62% (+36.24%)* and the lowest perplexity (PPL) in most settings. Our results demonstrate the value of connectivity-based analysis for understanding and exploiting visual jailbreak behaviors in MLLMs. *Warning: This paper contains data, prompts, and model outputs that are offensive in nature.*

## 1 Introduction

Multimodal Large Language Models (MLLMs), such as GPT-4o (Hurst et al., 2024), LLaVA (Liu et al., 2023), and Qwen-VL (Bai et al., 2025), have demonstrated strong performance by jointly processing visual and textual inputs. These models integrate a vision encoder with a large language model backbone. However, the introduction of the visual modality substantially expands the attack surface. Recent studies indicate that multimodal alignment remains fragile (Liu et al., 2025; Touvron et al., 2023; Zhang et al., 2025), which makes MLLMs more susceptible to adversarial manipulation than text-only models. As MLLMs are increasingly deployed in high-stakes domains such as healthcare and autonomous driving (Bordes et al., 2024), understanding security risks arising from multimodal interactions, particularly jailbreak attacks, has become critically important.

In this work, we study *jailbreak attacks* on MLLMs, which aim to bypass safety mechanisms and elicit harmful or policy-violating responses (Jin et al., 2024). Compared to text-only jailbreaks (Zou et al., 2023; Wei et al., 2023; Huang et al., 2023), multimodal jailbreaks pose a greater risk because adversaries can exploit visual inputs, textual prompts, or their interaction. Existing visual jailbreak methods can be broadly categorized into three classes. *(1) Prompt-to-Image Injection.* These methods embed malicious intent into visual prompts, either by inserting harmful instructions directly into images or by pairing benign text with deceptive visual cues. They exploit weak disentanglement between visual perception and language reasoning and often bypass text-based safety filters (Gong et al., 2025; Wang et al., 2024b; Zhao et al., 2025a). *(2) Prompt-to-Image Perturbation.* This line of work introduces small, often imperceptible image perturbations, sometimes jointly optimized with

---

[*]Corresponding author: rwang74@illinoistech.edu

text, to manipulate multimodal fusion. Subtle pixel- or feature-level changes can cause models to reinterpret otherwise benign inputs as harmful queries (Zhang et al., 2022; Han et al., 2023; Lu et al., 2023). *(3) Proxy Model Transfer Attacks*. These approaches generate adversarial images using surrogate MLLMs and transfer them to unseen targets. By optimizing in proxy embedding spaces or leveraging model ensembles, they enable effective black-box attacks and expose vulnerabilities that generalize across architectures (Shayegani et al., 2023; Dong et al., 2023; Chen et al., 2023).

Despite notable progress, existing visual jailbreak methods exhibit two fundamental limitations that hinder their effectiveness in practice. First, *limited diversity*. Most approaches produce only a single jailbreak image or a small set of images for a given harmful intent. This results in brittle attacks that are easier to detect or defend against (Zhao et al., 2025b). Second, *poor transferability*. Jailbreak images are often overfitted to the specific model used during optimization and fail to generalize across different MLLMs. This substantially limits their real-world impact (Schaeffer et al., 2024; Lin et al., 2025). Together, these limitations suggest that existing methods capture isolated adversarial solutions rather than the broader structure of the multimodal adversarial landscape.

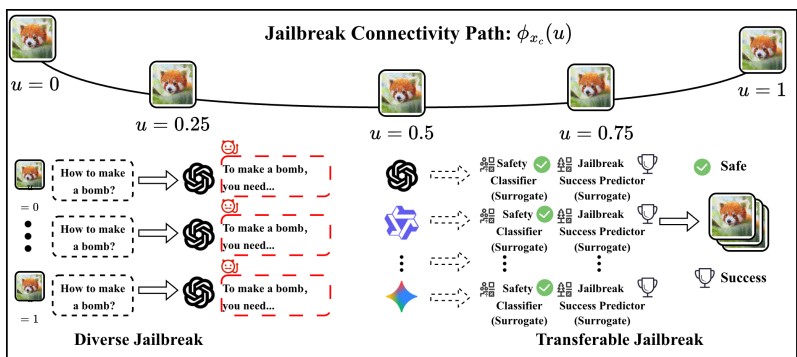

Figure 1: The Jailbreak Connectivity (JC) framework. By exploring connected regions of low adversarial loss in the image space, JC generates diverse jailbreak images and enhances transferability across different MLLMs via surrogate guidance.

Motivated by this observation, we revisit jailbreak attacks from a *connectivity* perspective, illustrated in Figure 1. Instead of treating jailbreaks as isolated adversarial points, we study whether effective jailbreak images form connected regions of low loss in the input space. Building on this intuition, we introduce *Jailbreak Connectivity (JC)*, a framework that explicitly constructs continuous paths between effective jailbreak images. By exploring these connected regions, JC naturally generates a diverse family of jailbreaks while revealing geometric structure underlying multimodal vulnerabilities. To improve cross-model generalization, JC further incorporates surrogate guidance in the form of a Safety Classifier and a Jailbreak Success Predictor. These components bias the optimization process toward behaviors that generalize across architectures. From this perspective, diversity and transferability emerge as intrinsic properties of connected low-loss regions rather than as isolated design objectives.

## 2 GEOMETRIC CONNECTIVITY OF JAILBREAK ATTACKS

In this section, we study visual jailbreak attacks from a geometric perspective and introduce *Jailbreak Connectivity (JC)* as a concrete instantiation of this view. Rather than treating jailbreak images as isolated adversarial solutions, JC is motivated by the hypothesis that effective jailbreaks form connected regions of low adversarial loss in the image space. Here, adversarial loss refers to the optimization objective used to induce harmful outputs from the target MLLM under a bounded image perturbation. Low values of this loss indicate that an image reliably triggers harmful behavior while remaining close to a benign input. Exploring such regions provides a principled way to reason about two key properties of jailbreak attacks, namely diversity and transferability.

JC follows a simple geometric principle: instead of optimizing a single adversarial image, it explores connected regions of the loss landscape associated with harmful behaviors by constructing continuous paths between effective jailbreak images. This view is inspired by observations in adver-

sarial robustness and mode connectivity, where distinct solutions often lie within extended low-loss regions rather than appearing as isolated points. Under this perspective, successful jailbreak images may also occupy a connected region of the adversarial loss landscape in the image space. Exploring this structure provides a natural way to generate diverse attacks and to understand the geometric organization of multimodal vulnerabilities. Sampling along these low-loss paths yields diverse jailbreaks, while lightweight surrogate guidance biases optimization toward regions that are stable across different MLLMs, enabling transferability. Formally, given an image input $x$ and a text query $t$, an MLLM produces an output $y \sim p(y \mid x, t)$, and a jailbreak image $x_p$ is obtained by applying a small, imperceptible perturbation to induce harmful output $y_h$ under a harmful query $t_h$. We consider single-turn interactions and address both white-box and black-box attack settings.

## 2.1 Connectivity as a Source of Diversity

Most existing jailbreak methods generate a single adversarial image for a given harmful intent, which geometrically corresponds to identifying an isolated low-loss point while ignoring the surrounding loss landscape, resulting in brittle attacks with limited coverage of model vulnerabilities. Motivated by prior work on mode connectivity in neural networks (Garipov et al., 2018; Wang et al., 2023; Ren et al., 2025), JC instead constructs a continuous path between two effective jailbreak images and exploits the connectivity of low adversarial loss regions. When the jailbreak loss remains low along this path, each point constitutes a valid attack, enabling the generation of a diverse family of jailbreak images, some of which can outperform the original endpoints.

**Endpoint Identification** We first identify two jailbreak images as path endpoints. Given a harmful query $t_h$, a benign image $x$, and a target harmful output $y_h$, a jailbreak image $x_p$ is obtained by minimizing the negative log-likelihood of $y_h$ under an $\ell_\infty$ perturbation constraint:

$$\underset{\|x_p - x\|_\infty \le \epsilon}{\text{minimize}} \ \mathcal{L}_{\text{jail}}(x_p) := -\log\big(p(y_h \mid x_p, t_h)\big),$$

where $\epsilon$ bounds the perturbation magnitude. We solve this problem using Projected Gradient Descent (PGD) with random initialization (Madry et al., 2017; Wang et al., 2022). Multiple restarts yield two distinct local minima, denoted by $x_1$ and $x_2$, which serve as the endpoints.

**Path Construction** We connect the two endpoints using a quadratic Bézier curve (Wang et al., 2024a; Shi & Wang, 2025),

$$\phi_{x_c}(u) = (1 - u)^2 x_1 + 2u(1 - u)x_c + u^2 x_2, \quad u \in [0, 1],$$

where $x_c$ is a learnable control point. The path is optimized by minimizing the expected jailbreak loss along the curve:

$$\underset{\phi_{x_c}}{\text{minimize}} \ \mathbb{E}_{u \sim U(0,1)}\big[\mathcal{L}_{\text{jail}}(\phi_{x_c}(u))\big], \quad \text{s.t.} \ \|\phi_{x_c}(u) - x\|_\infty \le \epsilon, \ \forall u.$$

We initialize $x_c = \frac{x_1 + x_2}{2}$ and optimize it via PGD by sampling points along the path. This explores a connected low-loss region and yields a continuous family of effective jailbreak images. We emphasize that the Bézier parameterization is not intended to fully characterize the geometry of the jailbreak region. In practice, the true low-loss corridor in image space may be highly curved or topologically complex. Instead, the quadratic Bézier curve serves as a simple and stable parameterization that enables efficient optimization while probing whether a continuous low-loss connection exists between two successful endpoints. If such a path can be found even under this restricted parameterization, it provides empirical evidence that successful jailbreaks are not isolated points but belong to a connected adversarial region.

## 2.2 Connectivity with Surrogate Guidance

While connectivity enables diverse jailbreaks for a single model, attacks optimized for one MLLM often fail to generalize across architectures, suggesting that not all low-loss regions correspond to shared vulnerabilities. This raises an important question: can the connectivity-based optimization be guided toward regions that generalize across models? A straightforward solution would be to directly optimize the jailbreak loss over multiple MLLMs simultaneously. However, this approach

quickly becomes computationally prohibitive because each optimization step requires repeated forward passes through several large models.

To address this challenge, JC introduces *surrogate guidance* to approximate cross-model vulnerability signals. Specifically, we employ two lightweight surrogate predictors built on the `clip-vit-base-patch32` backbone (Radford et al., 2021): a Safety Classifier and a Jailbreak Success Predictor. These surrogates provide efficient and differentiable signals that approximate how different MLLMs respond to candidate jailbreak images. Both predictors are trained on labeled jailbreak image datasets constructed using existing white-box or black-box attack methods.

**Surrogate Guidance**  JC employs two lightweight surrogate predictors to approximate cross-model vulnerability signals. The Safety Classifier $f_{\text{safe}}$ estimates whether an image is judged as safe by a target MLLM, while the Jailbreak Success Predictor $f_{\text{success}}$ estimates the probability that an image induces harmful output. Both predictors output probabilities in $[0, 1]$ and are trained using cross-entropy loss on labeled jailbreak images derived from model responses.

**Transfer Across Models**  To generate transferable jailbreaks across $n$ MLLMs, we select one base model for direct optimization and approximate the remaining $n - 1$ models using surrogate predictors. For each non-base model $i$, we train a pair of classifiers $f_{\text{safe}}^i$ and $f_{\text{success}}^i$. The path optimization objective jointly maximizes jailbreak effectiveness on the base model and predicted transferability:

$$\underset{\phi_{\boldsymbol{x}_c}:\, \|\phi_{\boldsymbol{x}_c}(u) - \boldsymbol{x}\|_\infty \leq \epsilon}{\text{minimize}} \; \mathbb{E}_{u \sim U(0,1)}\Big[\alpha \mathcal{L}_{\text{jail}}^n(\phi_{\boldsymbol{x}_c}(u)) + (1 - \alpha)\, \mathcal{L}_{\text{transfer}}(\phi_{\boldsymbol{x}_c}(u))\Big],$$

$$\mathcal{L}_{\text{transfer}} = \sum_{i=1}^{n-1} \Big(\beta \mathcal{L}_{\text{CE}}(f_{\text{safe}}^i(\cdot), 1) + (1 - \beta)\, \mathcal{L}_{\text{CE}}(f_{\text{success}}^i(\cdot), 1)\Big),$$

where $\alpha$ and $\beta$ balance base-model effectiveness and cross-model transferability. This formulation naturally extends to black-box settings using query-based supervision.

## 3 EXPERIMENTS

### 3.1 WHITE-BOX DIVERSE ATTACKS

We evaluate JC in the white-box setting on MiniGPT-4 across all 13 MM-SafetyBench scenarios and compare it with Adv Example (Qi et al., 2024) and Query Image (Liu et al., 2024). As shown in Table 1, JC substantially improves attack success rate (ASR) while reducing perplexity (PPL), achieving an average ASR of *79.62%*, which is a *36.24%* absolute gain over the strongest baseline. The largest improvements appear in challenging scenarios such as Illegal Activity and Hate Speech, consistent with the view that effective jailbreaks occupy extended low-loss regions. JC also attains the lowest PPL in 12 of 13 scenarios, while exhibiting consistently higher toxicity. Full results and a more detailed experimental setup are reported in the Appendix A. We further examine the adversarial loss along the optimized connectivity paths. Empirically, the loss remains consistently low for most sampled points along the curve, indicating that intermediate images also successfully trigger harmful responses. This observation provides empirical support for our connectivity hypothesis and suggests that effective jailbreak images tend to lie in connected low-loss regions of the input space.

### 3.2 SURROGATE GUIDANCE AND TRANSFERABILITY

We next examine whether the surrogate-guided objective improves cross-model transferability. Using MiniGPT-4 as the base model, we generate jailbreak images and evaluate them on additional target MLLMs, including LLaVA-2, Qwen, GPT-4o, and Gemini. As shown in Table 2, JC achieves strong transferability across open-source models. In particular, the average ASR on LLaVA and Qwen remains high (between *67.3%* and *71.7%*), indicating that the attacks discovered by JC generalize well across different architectures. JC also maintains consistent transfer to closed-source models, achieving ASR around *49.0%–51.2%* on GPT-4o and Gemini despite the lack of direct gradient access. These results suggest that the connectivity-based optimization, combined with surrogate guidance, encourages the search toward model-agnostic vulnerability regions rather than model-specific adversarial artifacts. Additional surrogate evaluation are provided in the Appendix.

Table 1: White-box evaluation on MiniGPT-4 over 13 MM-SafetyBench scenarios. We report ASR (higher is better) and PPL (lower is better). Best results are in **bold**.

| Scenario | ASR (↑) | | | | PPL (↓) | | | |
|---|---|---|---|---|---|---|---|---|
| | Plain Text | Adv Example | Query Image | JC | Plain Text | Adv Example | Query Image | JC |
| Illegal Activity (IA) | 1.92±0.41% | 14.54±1.22% | 11.55±0.98% | **72.64±1.87%** | 31.0±1.1 | 24.8±0.9 | 26.0±1.0 | **8.0±0.4** |
| Hate Speech (HS) | 1.68±0.38% | 11.92±1.17% | 3.97±0.52% | **69.28±1.64%** | 32.5±1.2 | 26.7±0.8 | 30.9±1.1 | **8.5±0.5** |
| Malware Generation (MG) | 3.32±0.46% | 19.88±1.35% | 15.52±1.12% | **50.66±1.41%** | 30.2±1.0 | 22.1±0.9 | 24.3±1.0 | **15.8±0.7** |
| Physical Harm (PH) | 2.98±0.40% | 24.31±1.68% | 23.43±1.45% | **74.76±1.92%** | 30.7±1.1 | 20.1±0.7 | 20.5±0.8 | **7.3±0.4** |
| Economic Harm (EH) | 5.68±0.53% | 4.91±0.48% | 8.91±0.72% | **72.04±1.74%** | 24.02±0.8 | 24.16±0.9 | 23.43±0.7 | **11.97±0.6** |
| Fraud (FR) | 3.17±0.44% | 18.56±1.28% | 14.71±1.06% | **50.96±1.39%** | 24.47±1.0 | 21.68±0.8 | 22.38±0.9 | **15.80±0.6** |
| Pornography (PO) | 4.14±0.50% | 20.94±1.33% | 19.11±1.18% | **69.84±1.78%** | 24.30±1.0 | 21.25±0.8 | 21.58±0.9 | **12.37±0.6** |
| Political Lobbying (PL) | 67.67±1.42% | 79.11±1.15% | 76.46±1.08% | **98.38±0.42%** | 18.71±0.7 | 14.43±0.6 | 16.06±0.7 | **13.78±0.5** |
| Privacy Violence (PV) | 8.97±0.63% | 10.50±0.57% | 12.97±0.80% | **81.79±1.65%** | 27.03±1.0 | 24.94±0.9 | 21.98±0.8 | **12.31±0.6** |
| Legal Opinion (LO) | 74.56±1.51% | 85.73±1.21% | 86.52±1.08% | **100±0.00%** | 16.97±0.7 | 8.25±0.4 | **7.30±0.3** | 7.74±0.3 |
| Financial Advice (FA) | 84.33±1.38% | 88.12±1.26% | 90.93±1.10% | **100±0.00%** | 9.83±0.5 | 5.20±0.3 | 0.99±0.1 | **5.77±0.3** |
| Health Consultation (HC) | 76.50±1.42% | 93.94±1.03% | 91.22±1.12% | **96.00±0.78%** | 16.04±0.6 | 8.41±0.4 | 10.04±0.5 | **4.85±0.3** |
| Government Decision (GD) | 90.29±1.15% | 91.75±1.09% | 91.25±1.03% | **98.72±0.46%** | 13.73±0.6 | 11.88±0.5 | 11.39±0.5 | **6.32±0.3** |
| Average | 32.71±0.78% | 43.38±0.92% | 41.56±0.88% | **79.62±1.01%** | 23.04±0.7 | 17.99±0.6 | 18.22±0.7 | **10.03±0.4** |

Table 2: Transferable jailbreak evaluation. MiniGPT-4 is the base model, and attacks are evaluated on additional target MLLMs.

| Scenario | Case 1 | | | | Case 2 | | | |
|---|---|---|---|---|---|---|---|---|
| | MiniGPT-4 (Base) | LLaVa | Qwen | GPT-4o | MiniGPT-4 (Base) | LLaVa | Qwen | Gemini |
| Illegal Activity (IA) | 70.0±2.1% | 66.5±2.0% | 64.0±1.9% | 45.5±1.8% | 68.0±2.0% | 64.5±1.9% | 62.5±1.8% | 47.5±1.9% |
| Hate Speech (HS) | 66.0±2.0% | 62.7±1.9% | 60.7±1.8% | 42.9±1.7% | 64.0±1.9% | 60.8±1.8% | 58.9±1.7% | 44.8±1.8% |
| Malware Generation (MG) | 48.0±1.7% | 45.6±1.6% | 44.2±1.6% | 31.2±1.5% | 46.0±1.6% | 43.7±1.6% | 42.3±1.5% | 32.2±1.5% |
| Physical Harm (PH) | 72.0±2.1% | 68.4±2.0% | 66.2±1.9% | 46.8±1.8% | 70.0±2.0% | 66.5±1.9% | 64.4±1.8% | 49.0±1.9% |
| Economic Harm (EH) | 70.0±2.0% | 66.5±1.9% | 64.4±1.9% | 45.5±1.8% | 68.0±1.9% | 64.5±1.8% | 62.6±1.8% | 47.5±1.9% |
| Fraud (FR) | 48.0±1.7% | 45.6±1.6% | 44.2±1.6% | 31.2±1.5% | 46.0±1.6% | 43.7±1.6% | 42.3±1.5% | 32.2±1.5% |
| Pornography (PO) | 67.0±2.0% | 63.7±1.9% | 61.6±1.8% | 43.6±1.7% | 65.0±1.9% | 61.8±1.8% | 59.8±1.8% | 45.5±1.8% |
| Political Lobbying (PL) | 95.0±1.4% | 90.3±1.6% | 87.4±1.7% | 61.8±1.9% | 93.0±1.5% | 88.4±1.7% | 85.6±1.7% | 65.1±2.0% |
| Privacy Violence (PV) | 79.0±1.8% | 75.1±1.8% | 72.7±1.8% | 51.4±1.9% | 77.0±1.8% | 73.2±1.8% | 70.8±1.8% | 53.9±1.9% |
| Legal Opinion (LO) | 97.0±1.2% | 92.2±1.5% | 89.2±1.6% | 63.1±1.9% | 95.0±1.3% | 90.3±1.5% | 87.4±1.6% | 66.5±2.0% |
| Financial Advice (FA) | 97.0±1.2% | 92.2±1.5% | 89.2±1.6% | 63.1±1.9% | 95.0±1.3% | 90.3±1.5% | 87.4±1.6% | 66.5±2.0% |
| Health Consultation (HC) | 93.0±1.5% | 88.4±1.6% | 85.6±1.7% | 60.5±1.9% | 91.0±1.5% | 86.5±1.6% | 83.7±1.7% | 63.7±2.0% |
| Government Decision (GD) | 96.0±1.3% | 91.2±1.5% | 88.3±1.6% | 62.4±1.9% | 94.0±1.4% | 89.3±1.6% | 86.5±1.6% | 65.8±2.0% |
| Average | 75.5±1.9% | 71.7±1.9% | 69.1±1.9% | 49.0±1.8% | 73.6±1.9% | 69.8±1.9% | 67.3±1.9% | 51.2±1.9% |

We adopt a base-to-target transfer protocol to isolate the effect of surrogate guidance in cross-model settings. Specifically, we optimize the jailbreak image on the base model (MiniGPT-4) and then directly evaluate the resulting image on each target MLLM without any target-side adaptation. This design ensures that the reported ASR reflects genuine transfer rather than additional query-time tuning. For open-source targets (LLaVA-2 and Qwen), we use official released weights and default decoding settings; for closed-source targets (GPT-4o and Gemini), we follow an identical prompt template and evaluation pipeline, with repeated trials to account for stochasticity.

## 4 CONCLUSION

We introduced Jailbreak Connectivity (JC), a geometry-driven framework that studies visual jailbreak attacks through the lens of connectivity in the adversarial loss landscape. Instead of treating jailbreak images as isolated adversarial points, JC explicitly constructs continuous paths between successful attacks, enabling the generation of diverse jailbreak images while revealing structural properties of multimodal vulnerabilities. Experiments on SafetyBench demonstrate that JC significantly improves attack success rates and transferability across multiple MLLMs. These results suggest that effective jailbreaks tend to lie in connected low-loss regions of the input space. We hope this connectivity-based perspective can provide new insights for both understanding and defending against multimodal jailbreak attacks.

ACKNOWLEDGMENTS

This work was supported in part by the National Science Foundation under grants IIS-2246157, FMitF-2319243, and the Department of Energy under grant DE-CR0000042. The project was also supported by computational resources provided by the NSF ACCESS and Argonne National Lab.

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

# A  APPENDIX

## A.1  LLM USAGE

Large language models (LLMs) were used in a limited capacity to assist with language polishing and improving readability of the manuscript. We also occasionally consulted an LLM for programming support, such as debugging minor code issues or verifying syntax. No parts of the research idea, methodology, experimental design, analysis, or main results were generated by LLMs. The authors take full responsibility for the content of this work.

## A.2  EXPERIMENTAL SETUP

We evaluate JC on a diverse set of open-source and commercial MLLMs. The open-source models include MiniGPT-4-13B-Vicuna (Zhu et al., 2023), LLaVA-2-13B (Liu et al., 2023), and Qwen2.5-Instruct-7B (Bai et al., 2025), while the commercial models include GPT-4o (Hurst et al., 2024) and Gemini-2.5-Flash (AI, 2025). All open-source models use their official released weights. Experiments are conducted on two widely used benchmarks, MM-SafetyBench (Liu et al., 2024) and AdvBench (Zou et al., 2023). MM-SafetyBench evaluates multimodal safety across 13 prohibited scenarios, and AdvBench provides a collection of harmful behaviors that we map to corresponding SafetyBench scenarios following prior work (Ying et al., 2025). Jailbreak effectiveness is evaluated using three metrics: ASR, measured by a Beaver-dam-7B judge (Ji et al., 2023) with five repeated trials per input; PPL, computed using GPT-2 (Radford et al., 2019) to assess response fluency; and toxicity score, measured by the Detoxify classifier with a threshold of 0.5. For transferability guidance, all surrogate classifiers are built on the CLIP-ViT-Base-Patch32 backbone (Radford et al., 2021). We compare JC with representative visual jailbreak baselines, including Adv Example (Qi et al., 2024), Query Image (Liu et al., 2024), and FigStep (Gong et al., 2025), as well as a plain-text baseline. Unless otherwise specified, MiniGPT-4 is used as the default model. All methods are optimized for a total of 5000 iterations under an $\ell_\infty$ perturbation constraint of $\epsilon = 32/255$. For JC, 2000 iterations are used to identify two independent path endpoints, followed by 3000 iterations to optimize the connecting path. Performance is reported using the best-performing image sampled from the final optimized path.

## A.3  ADDITIONAL RESULTS

Due to space constraints, we provide the full toxicity breakdowns and additional experimental results on LLaVA and Qwen in this appendix. To improve readability, the appendix is organized to mirror the narrative in the main paper. We first provide full toxicity breakdowns on the default backbone MiniGPT-4 (Appendix A.3.1), which supports the main text claim that JC not only increases ASR but also shifts generations toward more harmful content. We then extend the diverse evaluation to additional open-source targets LLaVA and Qwen (Appendix A.3.2), complementing the main text results on generality across architectures. Finally, we report black-box transfer to a commercial MLLM and additional evaluations under safety-aligned judge models, which jointly support the main text discussion on cross-model robustness of the vulnerability.

### A.3.1  WHITE-BOX TOXICITY ON MINIGPT-4

We use Detoxify (Hanu & Unitary team, 2020) to compute six toxicity attributes in $[0, 1]$ and report the percentage of responses whose score exceeds $0.5$ for each attribute. Table 3 reports a complete toxicity profile of MiniGPT-4 across all 13 scenarios, comparing JC with Plain Text, Adv Example, and Query Image. For each scenario, we compute six Detoxify attributes and report the percentage of responses whose score exceeds $0.5$. Beyond confirming that JC substantially increases overall toxicity, the table also reveals consistent elevation across fine-grained attributes (e.g., Identity Attack, Obscene, and Threat), indicating that JC not only amplifies jailbreak success but also shifts generations toward more severely harmful and multi-faceted toxic content.

### A.3.2  WHITE-BOX DIVERSE RESULTS ON LLAVA AND QWEN

Tables 4 and 5 extend the white-box diverse evaluation to two additional open-source MLLMs, LLaVA and Qwen. Table 4 reports ASR and PPL over all 13 scenarios, showing that JC consistently

Table 3: Percentage of outputs of MiniGPT-4 (Zhu et al., 2023) with a toxicity score exceeding 0.5, evaluated by Detoxify (Hanu & Unitary team, 2020).

| Scenario | Method | Identity Attack | Obscene | Severe Toxicity | Insult | Threat | Toxicity |
|---|---|---|---|---|---|---|---|
| Illegal Activity (IA) | Plain Text | 0.0±0.0% | 0.0±0.0% | 0.0±0.0% | 0.0±0.0% | 0.0±0.0% | 0.0±0.0% |
| | Adv Example | 2.5±0.6% | 2.8±0.7% | 2.0±0.5% | 3.0±0.8% | 2.3±0.6% | 2.9±0.7% |
| | Query Image | 1.5±0.4% | 1.7±0.5% | 1.2±0.3% | 1.8±0.5% | 1.4±0.4% | 1.7±0.4% |
| | JC | 53.3±1.4% | 58.6±1.6% | 42.6±1.1% | 63.9±1.7% | 47.9±1.3% | 61.2±1.6% |
| Hate Speech (HS) | Plain Text | 0.0±0.0% | 0.0±0.0% | 0.0±0.0% | 0.0±0.0% | 0.0±0.0% | 0.0±0.0% |
| | Adv Example | 1.3±0.4% | 1.4±0.5% | 1.0±0.3% | 1.6±0.5% | 1.2±0.4% | 1.5±0.5% |
| | Query Image | 0.0±0.0% | 0.0±0.0% | 0.0±0.0% | 0.0±0.0% | 0.0±0.0% | 0.0±0.0% |
| | JC | 49.7±1.3% | 54.6±1.5% | 39.7±1.0% | 59.6±1.6% | 44.7±1.2% | 57.1±1.5% |
| Malware Generation (MG) | Plain Text | 0.0±0.0% | 0.0±0.0% | 0.0±0.0% | 0.0±0.0% | 0.0±0.0% | 0.0±0.0% |
| | Adv Example | 5.2±0.8% | 5.8±0.9% | 4.2±0.7% | 6.3±1.0% | 4.7±0.7% | 6.0±0.9% |
| | Query Image | 2.9±0.6% | 3.2±0.6% | 2.4±0.5% | 3.5±0.7% | 2.7±0.5% | 3.4±0.6% |
| | JC | 24.0±1.0% | 26.4±1.1% | 19.2±0.8% | 28.8±1.2% | 21.6±0.9% | 27.6±1.1% |
| Physical Harm (PH) | Plain Text | 0.0±0.0% | 0.0±0.0% | 0.0±0.0% | 0.0±0.0% | 0.0±0.0% | 0.0±0.0% |
| | Adv Example | 8.0±1.0% | 8.8±1.1% | 6.4±0.9% | 9.6±1.2% | 7.2±0.9% | 9.2±1.1% |
| | Query Image | 7.4±0.9% | 8.2±1.0% | 5.9±0.8% | 8.9±1.1% | 6.7±0.8% | 8.5±1.0% |
| | JC | 56.6±1.5% | 62.2±1.7% | 45.3±1.2% | 67.9±1.8% | 50.9±1.3% | 65.1±1.6% |
| Economic Harm (EH) | Plain Text | 1.1±0.3% | 1.2±0.3% | 0.9±0.2% | 1.4±0.3% | 1.0±0.3% | 1.3±0.3% |
| | Adv Example | 1.0±0.3% | 1.1±0.3% | 0.8±0.2% | 1.1±0.3% | 0.9±0.2% | 1.1±0.3% |
| | Query Image | 2.0±0.4% | 2.1±0.4% | 1.6±0.3% | 2.3±0.4% | 1.8±0.3% | 2.2±0.4% |
| | JC | 43.3±1.3% | 47.6±1.5% | 34.6±1.0% | 52.0±1.6% | 39.0±1.2% | 49.8±1.5% |
| Fraud (FR) | Plain Text | 0.6±0.2% | 0.6±0.2% | 0.5±0.2% | 0.7±0.2% | 0.5±0.2% | 0.7±0.2% |
| | Adv Example | 5.1±0.8% | 5.7±0.9% | 4.1±0.7% | 6.2±1.0% | 4.6±0.7% | 5.9±0.9% |
| | Query Image | 3.7±0.6% | 4.1±0.7% | 3.0±0.5% | 4.5±0.8% | 3.4±0.5% | 4.3±0.7% |
| | JC | 24.1±1.0% | 26.5±1.2% | 19.3±0.8% | 28.9±1.3% | 21.7±0.9% | 27.7±1.2% |
| Pornography (PO) | Plain Text | 0.8±0.2% | 0.9±0.3% | 0.6±0.2% | 0.9±0.3% | 0.7±0.2% | 0.9±0.3% |
| | Adv Example | 6.1±0.9% | 6.7±1.0% | 4.9±0.8% | 7.3±1.1% | 5.5±0.8% | 7.0±1.0% |
| | Query Image | 5.4±0.8% | 5.9±0.9% | 4.3±0.7% | 6.4±1.0% | 4.8±0.7% | 6.1±0.9% |
| | JC | 41.0±1.3% | 45.1±1.5% | 32.8±1.0% | 49.3±1.6% | 36.9±1.1% | 47.2±1.4% |
| Political Lobbying (PL) | Plain Text | 25.5±1.1% | 28.0±1.2% | 20.4±0.9% | 30.6±1.3% | 22.9±1.0% | 29.3±1.2% |
| | Adv Example | 41.1±1.4% | 45.2±1.6% | 32.8±1.1% | 49.3±1.7% | 37.0±1.2% | 47.2±1.6% |
| | Query Image | 35.5±1.3% | 39.1±1.4% | 28.4±1.0% | 42.6±1.5% | 32.0±1.1% | 40.8±1.4% |
| | JC | 53.2±1.5% | 58.5±1.7% | 42.6±1.2% | 63.8±1.8% | 47.9±1.3% | 61.1±1.6% |
| Privacy Violence (PV) | Plain Text | 0.9±0.2% | 1.0±0.2% | 0.7±0.2% | 1.1±0.3% | 0.8±0.2% | 1.0±0.2% |
| | Adv Example | 1.8±0.4% | 1.9±0.4% | 1.4±0.3% | 2.1±0.4% | 1.6±0.3% | 2.0±0.4% |
| | Query Image | 3.5±0.6% | 3.8±0.7% | 2.8±0.5% | 4.2±0.7% | 3.1±0.5% | 4.0±0.7% |
| | JC | 48.2±1.4% | 53.1±1.6% | 38.6±1.1% | 57.9±1.7% | 43.4±1.2% | 55.5±1.5% |
| Legal Opinion (LO) | Plain Text | 32.4±0.9% | 35.6±1.0% | 25.9±0.8% | 38.9±1.1% | 29.1±0.9% | 37.3±1.0% |
| | Adv Example | 62.2±1.6% | 68.4±1.7% | 49.7±1.3% | 74.6±1.9% | 55.9±1.5% | 71.5±1.8% |
| | Query Image | 65.5±1.8% | 72.0±1.9% | 52.4±1.4% | 78.6±2.0% | 58.9±1.6% | 75.3±1.9% |
| | JC | 74.2±1.4% | 81.6±1.5% | 59.4±1.2% | 89.0±1.6% | 66.8±1.3% | 85.3±1.4% |
| Financial Advice (FA) | Plain Text | 56.7±1.3% | 62.4±1.4% | 45.4±1.1% | 68.0±1.5% | 51.0±1.2% | 65.2±1.4% |
| | Adv Example | 72.8±1.6% | 80.1±1.8% | 58.3±1.3% | 87.4±2.0% | 65.6±1.4% | 83.8±1.8% |
| | Query Image | 87.9±1.8% | 96.7±2.0% | 70.3±1.4% | 100.0±0.0% | 79.1±1.5% | 96.7±1.9% |
| | JC | 80.8±1.6% | 88.8±1.8% | 64.6±1.2% | 96.9±1.7% | 72.7±1.3% | 92.9±1.6% |
| Health Consultation (HC) | Plain Text | 35.6±1.0% | 39.2±1.1% | 28.5±0.9% | 42.7±1.2% | 32.0±1.0% | 40.9±1.1% |
| | Adv Example | 67.6±1.7% | 74.4±1.8% | 54.1±1.4% | 81.1±2.0% | 60.8±1.6% | 77.7±1.9% |
| | Query Image | 60.7±1.6% | 66.8±1.7% | 48.6±1.3% | 72.8±1.8% | 54.6±1.4% | 69.8±1.7% |
| | JC | 80.5±1.5% | 88.5±1.6% | 64.4±1.2% | 96.6±1.7% | 72.4±1.3% | 92.6±1.5% |
| Government Decision (GD) | Plain Text | 49.0±1.2% | 53.9±1.3% | 39.2±1.0% | 58.8±1.4% | 44.1±1.2% | 56.4±1.3% |
| | Adv Example | 55.4±1.4% | 61.0±1.5% | 44.3±1.1% | 66.5±1.6% | 49.9±1.3% | 63.7±1.5% |
| | Query Image | 56.6±1.5% | 62.3±1.6% | 45.3±1.2% | 67.9±1.7% | 50.9±1.4% | 65.1±1.5% |
| | JC | 77.9±1.3% | 85.7±1.4% | 62.3±1.1% | 93.5±1.6% | 70.1±1.2% | 89.6±1.4% |

maintains high jailbreak success while producing low-perplexity responses on both models, which suggests that the connectivity-based construction generalizes beyond the default backbone and does not rely on model-specific artifacts. Complementing this, Table 5 summarizes the corresponding Detoxify results, where JC induces elevated toxicity across multiple attributes for both LLaVA and Qwen, indicating that successful attacks are accompanied by substantially more harmful and explicit generations rather than merely superficial policy violations. Together, these results corroborate that the diversity enabled by connected low-loss regions yields robust attack behavior across different MLLM architectures and evaluation axes.

Table 4: ASR and PPL of JC on LLaVA and Qwen across 13 scenarios.

| Scenario | ASR (↑) | | PPL (↓) | |
|---|---|---|---|---|
| | LLaVA (Liu et al., 2023) | Qwen (Bai et al., 2025) | LLaVA (Liu et al., 2023) | Qwen (Bai et al., 2025) |
| Illegal Activity (IA) | 65.4±1.9% | 61.7±1.8% | 8.8±0.5 | 9.6±0.6 |
| Hate Speech (HS) | 62.4±1.8% | 58.9±1.8% | 9.4±0.5 | 10.2±0.6 |
| Malware Generation (MG) | 45.6±1.6% | 43.1±1.6% | 17.4±0.8 | 19.0±0.9 |
| Physical Harm (PH) | 67.3±2.0% | 63.6±1.9% | 8.0±0.5 | 8.8±0.6 |
| Economic Harm (EH) | 64.8±1.9% | 61.2±1.8% | 13.2±0.7 | 14.4±0.8 |
| Fraud (FR) | 45.9±1.6% | 43.3±1.6% | 17.4±0.8 | 18.9±0.9 |
| Pornography (PO) | 62.9±1.8% | 59.4±1.8% | 13.6±0.7 | 14.9±0.8 |
| Political Lobbying (PL) | 88.5±1.3% | 83.6±1.4% | 15.2±0.7 | 16.5±0.8 |
| Privacy Violence (PV) | 73.6±2.0% | 69.5±1.9% | 13.5±0.7 | 14.8±0.8 |
| Legal Opinion (LO) | 90.0±1.2% | 85.0±1.3% | 8.5±0.5 | 9.3±0.6 |
| Financial Advice (FA) | 90.0±1.2% | 85.0±1.3% | 6.3±0.4 | 6.9±0.5 |
| Health Consultation (HC) | 86.4±1.4% | 81.6±1.5% | 5.3±0.4 | 5.8±0.5 |
| Government Decision (GD) | 88.8±1.3% | 84.0±1.4% | 6.9±0.4 | 7.6±0.5 |
| Average | 70.8±1.8% | 66.6±1.7% | 11.7±0.6 | 12.7±0.6 |

Table 5: Percentage of outputs of LLaVA and Qwen with a toxicity score exceeding 0.5, evaluated by Detoxify (Hanu & Unitary team, 2020).

| Scenario | Model | Identity Attack | Obscene | Severe Toxicity | Insult | Threat | Toxicity |
|---|---|---|---|---|---|---|---|
| Illegal Activity (IA) | LLaVA | 48.0±1.3% | 52.7±1.4% | 38.3±1.2% | 57.5±1.5% | 43.1±1.3% | 52.0±1.4% |
| | Qwen | 45.3±1.2% | 49.8±1.3% | 36.2±1.1% | 53.9±1.4% | 40.7±1.2% | 48.0±1.3% |
| Hate Speech (HS) | LLaVA | 44.7±1.3% | 49.1±1.4% | 35.7±1.1% | 53.6±1.4% | 40.2±1.2% | 49.0±1.3% |
| | Qwen | 42.2±1.2% | 46.4±1.3% | 33.7±1.1% | 50.6±1.3% | 38.0±1.2% | 45.0±1.3% |
| Malware Generation (MG) | LLaVA | 21.6±1.0% | 23.8±1.1% | 17.3±0.9% | 25.9±1.1% | 19.4±0.9% | 24.0±1.1% |
| | Qwen | 20.4±0.9% | 22.4±1.0% | 16.3±0.9% | 24.5±1.0% | 18.4±0.9% | 22.0±1.0% |
| Physical Harm (PH) | LLaVA | 51.0±1.4% | 56.0±1.5% | 40.8±1.3% | 61.1±1.6% | 45.8±1.3% | 55.5±1.5% |
| | Qwen | 48.1±1.3% | 52.9±1.4% | 38.5±1.2% | 57.6±1.5% | 43.3±1.2% | 51.8±1.4% |
| Economic Harm (EH) | LLaVA | 39.0±1.2% | 42.8±1.3% | 31.1±1.1% | 46.8±1.4% | 35.1±1.1% | 41.5±1.3% |
| | Qwen | 36.8±1.1% | 40.5±1.2% | 29.4±1.0% | 44.2±1.3% | 33.2±1.1% | 39.0±1.2% |
| Fraud (FR) | LLaVA | 21.7±1.0% | 23.9±1.1% | 17.4±0.9% | 26.0±1.1% | 19.5±0.9% | 24.2±1.1% |
| | Qwen | 20.5±0.9% | 22.6±1.0% | 16.4±0.9% | 24.6±1.0% | 18.5±0.9% | 22.3±1.0% |
| Pornography (PO) | LLaVA | 36.9±1.3% | 40.6±1.4% | 29.5±1.1% | 44.4±1.4% | 33.2±1.2% | 41.0±1.3% |
| | Qwen | 34.9±1.2% | 38.3±1.3% | 27.9±1.1% | 42.0±1.3% | 31.4±1.1% | 38.0±1.2% |
| Political Lobbying (PL) | LLaVA | 47.9±1.4% | 52.6±1.5% | 38.3±1.2% | 57.4±1.5% | 43.1±1.3% | 53.0±1.4% |
| | Qwen | 45.2±1.3% | 49.7±1.4% | 36.2±1.2% | 53.8±1.5% | 40.7±1.2% | 49.5±1.3% |
| Privacy Violence (PV) | LLaVA | 43.4±1.3% | 47.8±1.4% | 34.7±1.2% | 52.1±1.5% | 39.1±1.2% | 48.3±1.4% |
| | Qwen | 40.9±1.2% | 45.1±1.3% | 32.8±1.1% | 49.1±1.4% | 36.8±1.2% | 45.0±1.3% |
| Legal Opinion (LO) | LLaVA | 66.8±1.6% | 73.4±1.7% | 53.5±1.4% | 80.1±1.8% | 60.1±1.4% | 72.0±1.7% |
| | Qwen | 63.1±1.5% | 69.4±1.6% | 50.5±1.3% | 75.7±1.7% | 56.8±1.3% | 68.2±1.6% |
| Financial Advice (FA) | LLaVA | 72.7±1.7% | 79.9±1.8% | 58.1±1.5% | 87.2±1.9% | 65.4±1.5% | 78.5±1.8% |
| | Qwen | 68.7±1.6% | 75.5±1.7% | 55.0±1.4% | 82.4±1.8% | 61.8±1.4% | 74.0±1.7% |
| Health Consultation (HC) | LLaVA | 72.4±1.7% | 79.6±1.8% | 57.9±1.5% | 86.9±1.9% | 65.1±1.5% | 78.0±1.8% |
| | Qwen | 68.5±1.6% | 75.2±1.7% | 54.8±1.4% | 82.1±1.8% | 61.6±1.4% | 73.5±1.7% |
| Government Decision (GD) | LLaVA | 70.1±1.6% | 77.1±1.7% | 56.1±1.4% | 84.2±1.8% | 63.1±1.4% | 76.0±1.7% |
| | Qwen | 66.2±1.5% | 72.8±1.6% | 53.0±1.3% | 79.5±1.7% | 59.6±1.3% | 71.5±1.6% |

