# OpenReview forum: "Geometry-Driven Diverse and Transferable Visual Attacks on Multimodal LLMs"
_ICLR.cc/2026/Workshop/GRaM — ICLR 2026 Workshop GRaM Poster_

### Official Review · Reviewer_dA4j · 2026-02-22
**Novel perspective on designing jailbreaking inputs for Multimodal LLMs**

**Rating:** 5
**Confidence:** 3

**Review:**

**Strength**:
- New perspective about generation of jailbreaking images to guarantee their diversity and transferability across models.
- Impressive results compared to existing jailbreaking input generation.

Although interesting, there are some (major) **weaknesses** which, if addressed, could definitely make the submission stronger:
- The paper must add examples where the proposed method can generate input that can jailbreak better than existing methods.
- The hypothesis that effective jailbreaks form connected regions of low adversarial loss is never empirically/conceptually proven.
- The *transferability* property arises due to the prior knowledge about the base and other models rather than a fundamental identification of the general image-text alignment.
- The path between two jailbreak images is achieved with *quadratic Bezier curve*:
  - why would this be the best option? what other path definitions were considered?
  - why will this guarantee a path that consists of low-adversarial loss points?
  - is there a unique path between two jailbreak images?
  - the two initial images are developed with a bit of inherent randomness, how susceptible is the method for the noise in initialization?
- The proposed method depends on two surrogate models - a safety classifier and a jailbreak success predictor. Since they are essential for navigating the adversarial space across multiple models, they must be evaluated, and there is currently no clear measure of how good these surrogate models are.

**Pmlr Suitability:**

NA

---

### Official Review · Reviewer_Q6ok · 2026-02-24
**Reviews**

**Rating:** 6
**Confidence:** 2

**Review:**

Summary:
This paper proposes a geometry-driven visual jailbreak framework that, instead of optimizing a single adversarial example, optimizes a continuous path in image space to generate a diverse set of attack samples. Concretely, the path is parameterized as a quadratic Bézier curve, where the control point is the main optimization variable. To improve cross-model transferability, the method further incorporates two lightweight surrogate predictors (a safety classifier and a jailbreak success predictor) to provide vulnerability-related guidance during optimization.

Strength:
- Modeling jailbreaks as a path/connected region rather than a single point is an interesting and intuitive direction for improving attack diversity.
- The main results (e.g., Table 1) show substantial improvements over prior baselines with a relatively simple formulation.
-The reported transferability results across multiple MLLMs (Table 2) are promising and suggest the approach may generalize beyond the base model.

Weakness:
- The current design uses a single quadratic Bézier path (one control point). It would be valuable to explore richer constructions of the attack region (e.g., higher-order curves, multiple control points/piecewise paths, or region-based formulations such as convex sets) and study how these affect diversity and robustness.
- The method relies on surrogate predictors during optimization. This introduces additional training requirements and may add bias or limit generality. It would be interesting to investigate alternatives that reduce or remove surrogate dependence while maintaining transferability.

The paper presents a simple yet novel connectivity-based formulation for visual jailbreaks and reports strong gains in both white-box performance and cross-model transfer. While the approach currently relies on surrogate guidance and uses a limited path parameterization, the core idea is clear and the empirical results are compelling for a short/tiny-paper contribution.

**Pmlr Suitability:**

NA

---

### Official Review · Reviewer_SyRC · 2026-02-24

**Rating:** 5
**Confidence:** 3

**Review:**

**Overview**: The paper proposes Jailbreak Connectivity that models jailbreak images as the quadratic Bézier curve connecting two endpoints of jailbreak images, optimized to have low expected jailbreak loss. The paper also introduced Surrogate Guidance to transfer the jailbreak images across different models.

**Strengths**：
1. The paper introduces a geometric perspective, shifting the focus from finding isolated adversarial images to modeling effective jailbreaks as connected regions of low adversarial loss, which is novel and delicate on its own. This also ensures built-in attack diversity, addressing the brittleness of prior methods.
2. Empirically strong gains on the chosen benchmark: Main results show sizable ASR improvements and nontrivial transfer compared to prior baselines.


**Weakness**：
1. While the paper borrows a geometric concept to create JC, it generally does not elaborate on the motivation, intuition, and theoretical grounding of such design in terms of geometry. The core mechanism is more about connectivity itself, rather than the shape/geometry of the connecting path. This point is closely followed by point 2 and 3.
2. Limited expressiveness of the Bézier curve: The true “successful corridor” in image space may be highly curved and topologically complex. A single quadratic Bézier curve (with only one control point) may be too restrictive. More flexible parameterizations (e.g., piecewise Bézier curves, splines) could better capture such structure.
3. Connectivity hypothesis needs direct evidence: Using a Bézier parameterization can make it easier to find a continuous path between two successful endpoints, but this does not, by itself, establish that the underlying set of successful jailbreaks forms a genuinely connected low-loss region. Stronger theoretical basis and empirical validation would be needed to support the connectivity claim.

**Pmlr Suitability:**

NA

---

### Meta-Review · Area_Chair_AR1G · 2026-02-27

**Decision:**

Accept

**Metareview:**

Reviewers found the method interesting, and identified several key axis of improvement including increasing the expressiveness of the Bezier curve.

**Relevance To Proceedings:**

Tiny paper — does not apply

**Relevance To Workshop:**

Yes — suitable for GRaM

---

### Decision · Program_Chairs · 2026-03-02

Accept (Poster)